# Few-shot Video-to-Video Synthesis

**Ting-Chun Wang, Ming-Yu Liu, Andrew Tao, Guilin Liu, Jan Kautz, Bryan Catanzaro**

NVIDIA Corporation

`{tingchunw,mingyul,atao,guilinl,jkautz,bcatanzaro}@nvidia.com`

## Abstract

Video-to-video synthesis (`vid2vid`) aims at converting an input semantic video, such as videos of human poses or segmentation masks, to an output photorealistic video. While the state-of-the-art of `vid2vid` has advanced significantly, existing approaches share two major limitations. First, they are data-hungry. Numerous images of a target human subject or a scene are required for training. Second, a learned model has limited generalization capability. A pose-to-human `vid2vid` model can only synthesize poses of the single person in the training set. It does not generalize to other humans that are not in the training set. To address the limitations, we propose a *few-shot* `vid2vid` framework, which learns to synthesize videos of previously unseen subjects or scenes by leveraging few example images of the target at test time. Our model achieves this *few-shot generalization* capability via a novel network weight generation module utilizing an attention mechanism. We conduct extensive experimental validations with comparisons to strong baselines using several large-scale video datasets including human-dancing videos, talking-head videos, and street-scene videos. The experimental results verify the effectiveness of the proposed framework in addressing the two limitations of existing `vid2vid` approaches. Code is available at our [website](#).

## 1  Introduction

Video-to-video synthesis (`vid2vid`) refers to the task of converting an input semantic video to an output photorealistic video. It has a wide range of applications, including generating a human-dancing video using a human pose sequence [7, 12, 57, 67], or generating a driving video using a segmentation mask sequence [57]. Typically, to obtain such a model, one begins with collecting a training dataset for the target task. It could be a set of videos of a target person performing diverse actions or a set of street-scene videos captured by using a camera mounted on a car driving in a city. The dataset is then used to train a model that converts *novel* input semantic videos to corresponding photorealistic videos at test time. In other words, we expect a `vid2vid` model for humans can generate videos of the same person performing novel actions that are not in the training set and a street-scene `vid2vid` model can videos of novel street-scenes with the same style as those in the training set. With the advance of the generative adversarial networks (GANs) framework [13] and its image-conditional extensions [22, 58], existing `vid2vid` approaches have shown promising results.

We argue that generalizing to novel input semantic videos is insufficient. One should also aim for a model that can *generalize to unseen domains*, such as generating videos of human subjects that are not included in the training dataset. More ideally, a `vid2vid` model should be able to synthesize videos of unseen domains by leveraging just a few example images given at test time. If a `vid2vid` model cannot generalize to unseen persons or scene styles, then we must train a model for each new subject or scene style. Moreover, if a `vid2vid` model cannot achieve this domain generalization capability with only a few example images, then one has to collect many images for each new subject or scene style. This would make the model not easily scalable. Unfortunately, existing `vid2vid` approaches suffer from these drawbacks as they do not consider such generalization.

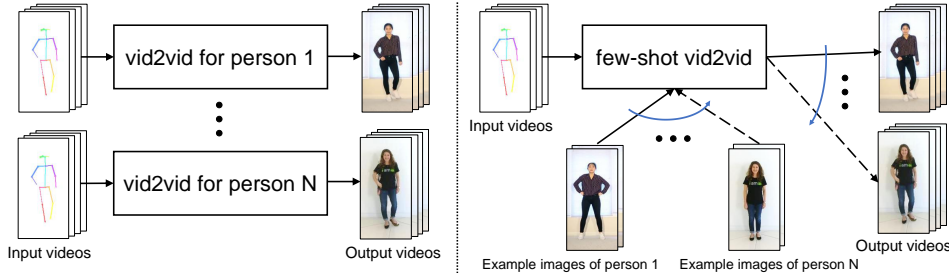

Figure 1: Comparison between the `vid2vid` (left) and the proposed `few-shot vid2vid` (right). Existing `vid2vid` methods [7, 12, 57] do not consider generalization to unseen domains. A trained model can only be used to synthesize videos similar to those in the training set. For example, a `vid2vid` model can only be used to generate videos of the person in the training set. To synthesize a new person, one needs to collect a dataset of the new person and uses it to train a new `vid2vid` model. On the other hand, our `few-shot vid2vid` model does not have the limitations. Our model can synthesize videos of new persons by leveraging few example images provided at the test time.

To address these limitations, we propose the *few-shot* `vid2vid` framework. The few-shot `vid2vid` framework takes two inputs for generating a video, as shown in Figure 1. In addition to the input semantic video as in `vid2vid`, it takes a second input, which consists of a few example images of the target domain made available at test time. Note that this is absent in existing `vid2vid` approaches [7, 12, 57, 67]. Our model uses these few example images to dynamically configure the video synthesis mechanism via a novel network weight generation mechanism. Specifically, we train a module to generate the network weights using the example images. We carefully design the learning objective function to facilitate learning the network weight generation module.

We conduct extensive experimental validation with comparisons to various baseline approaches using several large-scale video datasets including dance videos, talking head videos, and street-scene videos. The experimental results show that the proposed approach effectively addresses the limitations of existing `vid2vid` frameworks. Moreover, we show that the performance of our model is positively correlated with the diversity of the videos in the training dataset, as well as the number of example images available at test time. When the model sees more different domains in the training time, it can better generalize to deal with unseen domains (Figure 7(a)). When giving the model more example images at test time, the quality of synthesized videos improves (Figure 7(b)).

## 2 Related Work

**GANs.** The proposed `few-shot vid2vid` model is based on GANs [13]. Specifically, we use a conditional GAN framework. Instead of generating outputs by converting samples from some noise distribution [13, 42, 32, 14, 25], we generate outputs based on user input data, which allows more flexible control over the outputs. The user input data can take various forms, including images [22, 68, 30, 41], categorical labels [39, 35, 65, 4], textual descriptions [43, 66, 62], and videos [7, 12, 57, 67]. Our model belongs to the last one. However, different from the existing video-conditional GANs, which take the video as the sole data input, our model also takes a set of example images. These example images are provided at test time, and we use them to dynamically determine the network weights of our video synthesis model through a novel network weight generation module. This helps the network generate videos of unseen domains.

**Image-to-image synthesis**, which transfers an input image from one domain to a corresponding image in another domain [22, 50, 3, 46, 68, 30, 21, 69, 58, 8, 41, 31, 2], is the foundation of `vid2vid`. For videos, the new challenge lies in generating sequences of frames that are not only photorealistic individually but also temporally consistent as a whole. Recently, the FUNIT [31] was proposed for generating images of unseen domains via the adaptive instance normalization technique [19]. Our work is different in that we aim for video synthesis and achieve generalization to unseen domains via a network weight generation scheme. We compare these techniques in the experiment section.

**Video generative models** can be divided into three main categories, including 1) unconditional video synthesis models [54, 45, 51], which convert random noise samples to video clips, 2) future video prediction models [48, 24, 11, 34, 33, 63, 55, 56, 10, 53, 29, 27, 18, 28, 16, 40], which generate future video frames based on the observed ones, and 3) `vid2vid` models [57, 7, 12, 67], which convert semantic input videos to photorealistic videos. Our work belongs to the last category, but

in contrast to the prior works, we aim for a `vid2vid` model that can synthesize videos of unseen domains by leveraging few example images given at test time.

**Adaptive networks** refer to networks where part of the weights are dynamically computed based on the input data. This class of networks has a different inductive bias to regular networks and has found use in several tasks including sequence modeling [15], image filtering [23, 59, 49], frame interpolation [38, 37], and neural architecture search [64]. Here, we apply it to the `vid2vid` task.

**Human pose transfer** synthesizes a human in an unseen pose by utilizing an image of the human in a different pose. To achieve high quality generation results, existing human pose transfer methods largely utilize human body priors such as body part modeling [1] or human surface-based coordinate mapping [36]. Our work differs from these works in that our method is more general. We do not use specific human body priors other than the input semantic video. As a result, the same model can be directly used for other `vid2vid` tasks such as street scene video synthesis, as shown in Figure 5. Moreover, our model is designed for video synthesis, while existing human pose transfer methods are mostly designed for still image synthesis and do not consider the temporal aspect of the problem. As a result, our method renders more temporally consistent results (Figure 4).

## 3  Few-shot Video-to-Video Synthesis

Video-to-video synthesis aims at learning a mapping function that can convert a sequence of input semantic images[1], $\mathbf{s}_1^T \equiv \mathbf{s}_1, \mathbf{s}_2, ..., \mathbf{s}_T$, to a sequence of output images, $\tilde{\mathbf{x}}_1^T \equiv \tilde{\mathbf{x}}_1, \tilde{\mathbf{x}}_2, ..., \tilde{\mathbf{x}}_T$, in a way that the conditional distribution of $\tilde{\mathbf{x}}_1^T$ given $\mathbf{s}_1^T$ is similar to the conditional distribution of the ground truth image sequence, $\mathbf{x}_1^T \equiv \mathbf{x}_1, \mathbf{x}_2, ..., \mathbf{x}_T$, given $\mathbf{s}_1^T$. In other words, it aims to achieve $\mathcal{D}(p(\tilde{\mathbf{x}}_1^T|\mathbf{s}_1^T), p(\mathbf{x}_1^T|\mathbf{s}_1^T)) \rightarrow 0$, where $\mathcal{D}$ is a distribution divergence measure such as the Jensen-Shannon divergence or the Wasserstein distance. To model the conditional distribution, existing works make a simplified Markov assumption, leading to a sequential generative model given by

$$\tilde{\mathbf{x}}_t = F(\tilde{\mathbf{x}}_{t-\tau}^{t-1}, \mathbf{s}_{t-\tau}^t) \tag{1}$$

In other words, it generates the output image, $\tilde{\mathbf{x}}_t$, based on the observed $\tau + 1$ input semantic images, $\mathbf{s}_{t-\tau}^t$, and the past $\tau$ generated images, $\tilde{\mathbf{x}}_{t-\tau}^{t-1}$. The sequential generator $F$ can be modeled in several different ways [7, 12, 57, 67]. A popular choice is to use an image matting function given by

$$F(\tilde{\mathbf{x}}_{t-\tau}^{t-1}, \mathbf{s}_{t-\tau}^t) = (\mathbf{1} - \tilde{\mathbf{m}}_t) \odot \tilde{\mathbf{w}}_{t-1}(\tilde{\mathbf{x}}_{t-1}) + \tilde{\mathbf{m}}_t \odot \tilde{\mathbf{h}}_t \tag{2}$$

where the symbol $\mathbf{1}$ is an image of all ones, $\odot$ is the element-wise product operator, $\tilde{\mathbf{m}}_t$ is a soft occlusion map, $\tilde{\mathbf{w}}_{t-1}$ is the optical flow from $t - 1$ to $t$, and $\tilde{\mathbf{h}}_t$ is a synthesized intermediate image.

Figure 2(a) visualizes the `vid2vid` architecture and the matting function, which shows the output image $\tilde{\mathbf{x}}_t$ is generated by combining the optical-flow warped version of the last generated image, $\tilde{\mathbf{w}}_{t-1}(\tilde{\mathbf{x}}_{t-1})$, and the synthesized intermediate image, $\tilde{\mathbf{h}}_t$. The soft occlusion map, $\tilde{\mathbf{m}}_t$, dictates how these two images are combined at each pixel location. Intuitively, if a pixel is observed in the previously generated frame, it would favor duplicating the pixel value from the warped image. In practice, these quantities are generated via neural network-parameterized functions $M$, $W$, and $H$:

$$\tilde{\mathbf{m}}_t = M_{\boldsymbol{\theta}_M}(\tilde{\mathbf{x}}_{t-\tau}^{t-1}, \mathbf{s}_{t-\tau}^t), \tag{3}$$

$$\tilde{\mathbf{w}}_{t-1} = W_{\boldsymbol{\theta}_W}(\tilde{\mathbf{x}}_{t-\tau}^{t-1}, \mathbf{s}_{t-\tau}^t), \tag{4}$$

$$\tilde{\mathbf{h}}_t = H_{\boldsymbol{\theta}_H}(\tilde{\mathbf{x}}_{t-\tau}^{t-1}, \mathbf{s}_{t-\tau}^t) \tag{5}$$

where $\boldsymbol{\theta}_M$, $\boldsymbol{\theta}_W$, and $\boldsymbol{\theta}_H$ are learnable parameters. They are kept fixed once the training is done.

**Few-shot vid2vid.** While the sequential generator in (1) is trained for converting novel input semantic videos, it is not trained for synthesizing videos of unseen domains. For example, a model trained for a particular person can only be used to generate videos of the same person. In order to adapt $F$ to unseen domains, we let $F$ depend on extra inputs. Specifically, we let $F$ take two more input arguments: one is a set of $K$ example images $\{\mathbf{e}_1, \mathbf{e}_2, ..., \mathbf{e}_K\}$ of the target domain, and the other is the set of their corresponding semantic images $\{\mathbf{s}_{\mathbf{e}_1}, \mathbf{s}_{\mathbf{e}_2}, ..., \mathbf{s}_{\mathbf{e}_K}\}$. That is

$$\tilde{\mathbf{x}}_t = F(\tilde{\mathbf{x}}_{t-\tau}^{t-1}, \mathbf{s}_{t-\tau}^t, \{\mathbf{e}_1, \mathbf{e}_2, ..., \mathbf{e}_K\}, \{\mathbf{s}_{\mathbf{e}_1}, \mathbf{s}_{\mathbf{e}_2}, ..., \mathbf{s}_{\mathbf{e}_K}\}). \tag{6}$$

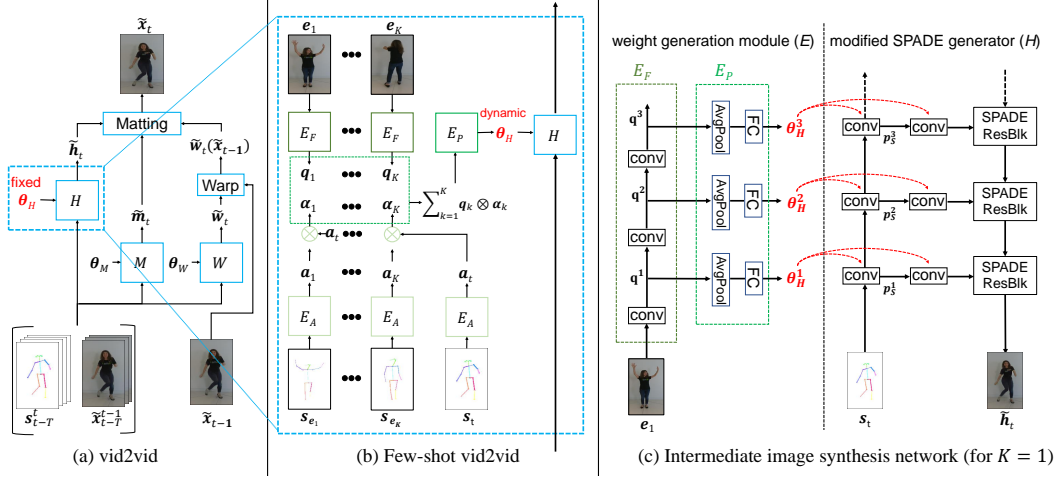

(a) vid2vid  (b) Few-shot vid2vid  (c) Intermediate image synthesis network (for $K = 1$)

Figure 2: (a) Architecture of the `vid2vid` framework [57]. (b) Architecture of the proposed few-shot `vid2vid` framework. It consists of a network weight generation module $E$ that maps example images to part of the network weights for video synthesis. The module $E$ consists of three sub-networks: $E_F$, $E_P$, and $E_A$ (used when $K > 1$). The sub-network $E_F$ extracts features $\mathbf{q}$ from the example images. When there are multiple example images ($K > 1$), $E_A$ combines the extracted features by estimating soft attention maps $\boldsymbol{\alpha}$ and weighted averaging different extracted features. The final representation is then fed into the network $E_P$ to generate the weights $\boldsymbol{\theta}_H$ for the image synthesis network $H$.

This modeling allows $F$ to leverage the example images given at the test time to extract some useful patterns to synthesize videos of the unseen domain. We propose a network weight generation module $E$ for extracting the patterns. Specifically, $E$ is designed to extract patterns from the provided example images and use them to compute network weights $\boldsymbol{\theta}_H$ for the intermediate image synthesis network $H$:

$$\boldsymbol{\theta}_H = E(\tilde{\mathbf{x}}_{t-\tau}^{t-1}, \mathbf{s}_{t-\tau}^t, \{\mathbf{e}_1, \mathbf{e}_2, ..., \mathbf{e}_K\}, \{\mathbf{s}_{\mathbf{e}_1}, \mathbf{s}_{\mathbf{e}_2}, ..., \mathbf{s}_{\mathbf{e}_K}\}). \tag{7}$$

Note that the network $E$ does not generate the weights $\boldsymbol{\theta}_M$ or $\boldsymbol{\theta}_W$ because the flow prediction network $W$ and the soft occlusion map prediction network $W$ are designed for warping the last generated image, and warping is a mechanism that is naturally shared across domains.

We build our `few-shot vid2vid` framework based on Wang *et al.* [57], which is the state-of-the-art for the `vid2vid` task. Specifically, we reuse their proposed flow prediction network $W$ and the soft occlusion map prediction network $M$. The intermediate image synthesis network $H$ is a conditional image generator. Instead of reusing the architecture proposed by Wang *et al.* [57], we adopt the SPADE generator [41], which is the current state-of-the-art semantic image synthesis model.

The SPADE generator contains several spatial modulation branches and a main image synthesis branch. Our network weight generation module $E$ only generates the weights for the spatial modulation branches. This has two main advantages. First, it greatly reduces the number of parameters that $E$ has to generate, which avoids the overfitting problem. Second, it avoids creating a shortcut from the example images to the output image, since the generated weights are only used in the spatial modulation modules, which generates the modulation values for the main image synthesis branch. In the following, we discuss details of the design of the network $E$ and the learning objective.

**Network weight generation module.** As discussed above, the goal of the network weight generation module $E$ is to learn to extract appearance patterns that can be injected into the video synthesis branch by controlling its weights. We first consider the case where only one example image is available ($K = 1$). We then extend the discussion to handle the case of multiple example images.

We decompose $E$ into two sub-networks: an example feature extractor $E_F$, and a multi-layer perceptron $E_P$. The network $E_F$ consists of several convolutional layers and is applied on the example image $\mathbf{e}_1$ to extract an appearance representation $\mathbf{q}$. The representation $\mathbf{q}$ is then fed into $E_P$ to generate the weights $\boldsymbol{\theta}_H$ in the intermediate image synthesis network $H$.

Let the image synthesis network $H$ has $L$ layers $H^l$, where $l \in [1, L]$. We design the weight generation network $E$ to also have $L$ layers, each $E^l$ generates the weights for the corresponding $H^l$. Specifically, to generate the weights $\boldsymbol{\theta}_H^l$ for layer $H^l$, we first take the output $\mathbf{q}^l$ from $l$-th layer in

$E_F$. Then, we average pool $\mathbf{q}^l$ (since $\mathbf{q}^l$ might be still a feature map with spatial dimensions.) and apply a multi-layer perceptron $E_P^l$ to generate the weights $\boldsymbol{\theta}_H^l$. Mathematically, if we define $\mathbf{q}^0 \equiv \mathbf{e}_1$, then $\mathbf{q}^l = E_F^l(\mathbf{q}^{l-1})$, and $\boldsymbol{\theta}_H^l = E_P^l(\mathbf{q}^l)$. These generated weights are then used to convolve the current input semantic map $\mathbf{s}_t$ to generate the normalization parameters used in SPADE (Figure 2(c)).

For each layer in the main SPADE generator, we use $\boldsymbol{\theta}_H^l$ to compute the denormalization parameters $\boldsymbol{\gamma}^l$ and $\boldsymbol{\beta}^l$ to denormalize the input features. We note that, in the original SPADE module, the scale map $\boldsymbol{\gamma}^l$ and bias map $\boldsymbol{\beta}^l$ are generated by fixed weights operated on the input semantic map $\mathbf{s}_t$. In our setting, these maps are generated by dynamic weights, $\boldsymbol{\theta}_H^l$. Moreover, $\boldsymbol{\theta}_H^l$ contains three sets of weights: $\boldsymbol{\theta}_S^l$, $\boldsymbol{\theta}_{\boldsymbol{\gamma}}^l$ and $\boldsymbol{\theta}_{\boldsymbol{\beta}}^l$. $\boldsymbol{\theta}_S^l$ acts as a shared layer to extract common features, and $\boldsymbol{\theta}_{\boldsymbol{\gamma}}^l$ and $\boldsymbol{\theta}_{\boldsymbol{\beta}}^l$ take the output of $\boldsymbol{\theta}_S^l$ to generate $\boldsymbol{\gamma}^l$ and $\boldsymbol{\beta}^l$ maps, respectively. For each BatchNorm layer in $G^l$, we compute the denormalized features $\mathbf{p}_H^l$ from the normalized features $\hat{\mathbf{p}}_H^l$ by

$$\mathbf{p}_S^l = \begin{cases} \mathbf{s}_t, & \text{if } l = 0 \\ \sigma\big(\mathbf{p}_S^{l-1} \circledast \boldsymbol{\theta}_S^l\big), & \text{otherwise} \end{cases} \tag{8}$$

$$\boldsymbol{\gamma}^l = \mathbf{p}_S^l \circledast \boldsymbol{\theta}_{\boldsymbol{\gamma}}^l, \quad \boldsymbol{\beta}^l = \mathbf{p}_S^l \circledast \boldsymbol{\theta}_{\boldsymbol{\beta}}^l \tag{9}$$

$$\mathbf{p}_H^l = \boldsymbol{\gamma}^l \odot \hat{\mathbf{p}}_H^l + \boldsymbol{\beta}^l \tag{10}$$

where $\circledast$ stands for convolution, and $\sigma$ is the nonlinearity function.

**Attention-based aggregation** ($K > 1$). In addition, we want $E$ to be capable of extracting the patterns from an arbitrary number of example images. As different example images may carry different appearance patterns, and they have different degrees of relevance to different input images, we design an attention mechanism [61, 52] to aggregate the extracted appearance patterns $\mathbf{q}_1,...,\mathbf{q}_K$.

To this end, we construct a new attention network $E_A$, which consists of several fully convolutional layers. $E_A$ is applied to each of the semantic images of the example images, $\mathbf{s}_{\mathbf{e}_k}$. This results in a key vector $\boldsymbol{a}_k \in \mathbb{R}^{C \times N}$, where $C$ is the number of channels and $N = H \times W$ is the spatial dimension of the feature map. We also apply $E_A$ to the current input semantic image $\mathbf{s}_t$ to extract its key vector $\boldsymbol{a}_t \in \mathbb{R}^{C \times N}$. We then compute the attention weight $\boldsymbol{\alpha}_k \in \mathbb{R}^{N \times N}$ by taking the matrix product $\boldsymbol{\alpha}_k = (\boldsymbol{a}_k)^T \otimes \boldsymbol{a}_t$. The attention weights are then used to compute a weighted average of the appearance representation $\mathbf{q} = \sum_{k=1}^K \mathbf{q}_k \otimes \boldsymbol{\alpha}_k$, which is then fed into the multi-layer perceptron $E_P$ to generate the network weights (Figure 2(b)). This aggregation mechanism is helpful when different example images contain different parts of the subject. For example, when example images include both front and back of the target person, the attention maps can help capture corresponding body parts during synthesis (Figure 7(c)).

**Warping example images.** To ease the burden of the image synthesis network, we can also (optionally) warp the given example image and combine it with the intermediate synthesized output $\tilde{\mathbf{h}}_t$. Specifically, we make the model estimate an additional flow $\tilde{\mathbf{w}}_{e_t}$ and mask $\tilde{\mathbf{m}}_{e_t}$, which are used to warp the example image $\mathbf{e}_1$ to the current input semantics, similar to how we warp and combine with previous frames. The new intermediate image then becomes

$$\tilde{\mathbf{h}}_t' = (\mathbf{1} - \tilde{\mathbf{m}}_{e_t}) \odot \tilde{\mathbf{w}}_{e_t}(\mathbf{e}_1) + \tilde{\mathbf{m}}_{e_t} \odot \tilde{\mathbf{h}}_t \tag{11}$$

In the case of multiple example images, we pick $\mathbf{e}_1$ to be the image that has the largest similarity score to the current frame by looking at the attention weights $\boldsymbol{\alpha}$. In practice, we found this helpful when example and target images are similar in most regions, such as synthesizing poses where the background remains static.

**Training.** We use the same learning objective as in the `vid2vid` framework [57]. But instead of training the `vid2vid` model using data from one domain, we use data from multiple domains. In Figure 7(a), we show the performance of our `few-shot vid2vid` model is positively correlated with the number of domains included in the training dataset. This shows that our model can gain from increased visual experiences. Our framework is trained in the supervised setting where paired $\mathbf{s}_1^T$, and $\mathbf{x}_1^T$ are available. We train our model to convert $\mathbf{s}_1^T$ to $\mathbf{x}_1^T$ by using $K$ example images randomly sampled from $\mathbf{x}$. We adopt a progressive training technique, which gradually increases the length of training sequences. Initially, we set $T = 1$, which means the network only generates single frames. After that, we double the sequence length ($T$) for every few epochs.

**Inference.** At test time, our model can take an arbitrary number of example images. In Figure 7(b), we show that our performance is positively correlated with the number of example images. Moreover, we can also (optionally) finetune the network using the given example images to improve performance. Note that we only finetune the weight generation module $E$ and the intermediate image synthesis network $H$, and leave all parameters related to flow estimation ($\boldsymbol{\theta}_M$, $\boldsymbol{\theta}_H$) fixed. We found this can better preserve the person identity in the example image.

## 4   Experiments

**Implementation details.** Our training procedure follows the procedure from the `vid2vid` work [57]. We use the ADAM optimizer [26] with lr $= 0.0004$ and $(\beta_1, \beta_2) = (0.5, 0.999)$. Training was conducted using an NVIDIA DGX-1 machine with 8 32GB V100 GPUs.

**Datasets.** We adopt three video datasets to validate our method.

- **YouTube dancing videos.** It consists of $1,500$ dancing videos from YouTube. We divide them into a training set and a test set with no overlapping subjects. Each video is further divided into short clips of continuous motions. This yields about $15,000$ clips for training. At each iteration, we randomly pick a clip and select one or more frames in the same clip as the example images. At test time, both the example images and the input human poses are not seen during training.

- **Street-scene videos.** We use street-scene videos from three different geographical areas: 1) Germany, from the Cityscapes dataset [9], 2) Boston, collected using a dashcam, and 3) NYC, collected by a different dashcam. We apply a pretrained segmentation network [60] to get the segmentation maps. Again, during training, we randomly select one frame of the same area as the example image. At test time, in addition to the test set images from these three areas, we also test on the ApolloScape [20] and CamVid [5] datasets, which are not included in the training set.

- **Face videos.** We use the real videos in the FaceForensics dataset [44], which contains $854$ videos of news briefing from different reporters. We split the dataset into $704$ videos for training and $150$ videos for validation. We extract sketches from the input videos similar to `vid2vid`, and select one frame of the same video as the example image to convert sketches to face videos.

**Baselines.** Since no existing `vid2vid` method can adapt to unseen domains using few example images, we construct 3 strong baselines that consider different ways of achieving the target generalization capability. For the following comparisons and figures, all methods use 1 example image.

- **Encoder.** In this baseline approach, we encode the example images into a style vector and then decode the features using the image synthesis branch in our $H$ to generate $\tilde{\mathbf{h}}_t$.

- **ConcatStyle.** In this baseline approach, we also encode the example images into a style vector. However, instead of directly decoding the style vector using the image synthesis branch in our $H$, it concatenates the vector with each of the input semantic images to produce an augmented semantic input image. This image is then used as input to the spatial modulation branches in our $H$ for generating the intermediate image $\tilde{\mathbf{h}}_t$.

- **AdaIN**. In this baseline, we insert an AdaIN normalization layer after each spatial modulation layer in the image synthesis branch of $H$. We generate the AdaIN normalization parameters by feeding the example images to an encoder, similar to the FUNIT method [31].

In addition to these baselines, for the human synthesis task, we also compare our approach with the following methods using the pretrained models provided by the authors.

- **PoseWarp** [1] synthesizes humans in unseen poses using an example image. The idea is to assume each limb undergoes a similarity transformation. The final output image is obtained by combining all transformed limbs together.

- **MonkeyNet** [47] is proposed for transferring motions from a sequence to a still image. It first detects keypoints in the images, and then predicts their flows for warping the still image.

**Evaluation metrics.** We use the following metrics for quantitative evaluation.

- **Fréchet Inception Distance (FID)** [17] measures the distance between the distributions of real data and generated data. It is commonly used to quantify the fidelity of synthesized images.

- **Pose error.** We estimate the poses of the synthesized subjects using OpenPose [6]. This renders a set of joint locations for each video frame. We then compute the absolute error in pixels between

| Method | YouTube Dancing videos | | | Street Scene videos | | | |
|---|---|---|---|---|---|---|---|
| | Pose Error | FID | Human Pref. | Pixel Acc | mIoU | FID | Human Pref. |
| Encoder | 13.30 | 234.71 | 0.96 | 0.400 | 0.222 | 187.10 | 0.97 |
| ConcatStyle | 13.32 | 140.87 | 0.95 | 0.479 | 0.240 | 154.33 | 0.97 |
| AdaIN | 12.66 | 207.18 | 0.93 | 0.756 | 0.360 | 205.54 | 0.87 |
| PoseWarp [1] | 16.84 | 180.31 | 0.83 | N/A | N/A | N/A | N/A |
| MonkeyNet [47] | 13.73 | 260.77 | 0.93 | N/A | N/A | N/A | N/A |
| **Ours** | **6.01** | **80.44** | — | **0.831** | **0.408** | **144.24** | — |

Table 1: Our method outperforms existing pose transfer methods and our baselines for both dancing and street scene video synthesis tasks. For pose error and FID, lower is better. For pixel accuracy and mIoU, higher is better. The human preference score indicates the fraction of subjects favoring results synthesized by our method.

Figure 3: Visualization of human video synthesis results. Given the same pose video but different example images, our method synthesizes realistic videos of the subjects, who are not seen during training. *The figure is best viewed with Acrobat Reader. Click the image to play the video clip.*

the estimated pose and the original pose input to the model. The idea behind this metric is that if the image is well-synthesized, a well-trained human pose estimation network should be able to recover the original pose used to synthesize the image. We note similar ideas were used in evaluating image synthesis performance in several prior works [22, 58, 57].

- **Segmentation accuracy.** To evaluate the performance of street scene videos, we run a state-of-the-art street scene segmentation network on the result videos generated by all the competing methods. We then report the pixel accuracy and mean intersection-over-union (IoU) ratio. The idea of using segmentation accuracy as a performance metric follows the discussion of using the pose error as discussed above.

- **Human subjective score.** Finally, we use Amazon Mechanical Turk (AMT) to evaluate the quality of generated videos. We perform AB tests where we provide the user videos from

Figure 4: Comparisons against different baselines for human motion synthesis. Note that the competing methods either have many visible artifacts or completely fail to transfer the motion. *The figure is best viewed with Acrobat Reader. Click the image to play the video clip.*

Figure 5: Visualization of street scene video synthesis results. Our approach is able to synthesize videos that realistically reflect the style in the example images even if the style is not included in the training set. *The figure is best viewed with Acrobat Reader. Click the image to play the video clip.*

two different approaches and ask them to choose the one with better quality. For each pair of comparisons, we generate 100 clips, each of them viewed by 60 workers. Orders are randomized.

**Main results.** In Figure 3, we show results of using different example images when synthesizing humans. It can be seen that our method can successfully transfer motion to all the example images. Figure 4 shows comparisons of our approaches against other methods. It can be seen that other methods either generate obvious artifacts or fail to transfer the motion faithfully.

Figure 5 shows the results of synthesizing street scene videos with different example images. It can be seen that even with the same input segmentation map, our method can achieve different visual results using different example images.

Table 1 shows quantitative comparisons of both tasks against the other methods. It can be seen that our method consistently achieves better results than the others on all the performance metrics.

In Figure 6, we show results of using different example images when synthesizing faces. Our method can faithfully preserve the person identity while capturing the motion in the input videos.

Finally, to verify our hypothesis that a larger training dataset helps improve the quality of synthesized videos, we conduct an experiment where part of the dataset is held out during training. We vary the number of videos in the training set and plot the resulting performance in Figure 7(a). We find that the results support our hypothesis. We also evaluate whether having access to more example images

Figure 6: Visualization of face video synthesis results. Given the same input video but different example images, our method synthesizes realistic videos of the subjects, who are not seen during training. *The figure is best viewed with Acrobat Reader. Click the image to play the video clip.*

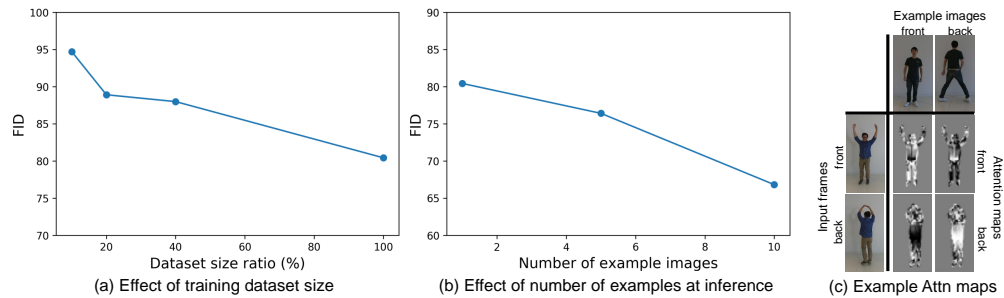

(a) Effect of training dataset size     (b) Effect of number of examples at inference     (c) Example Attn maps

Figure 7: (a) The plot shows the quality of our synthesized videos improves when it is trained with a larger dataset. Large variety helps learn a more generalizable network weight generation module and hence improves adaptation capability. (b) The plot shows the quality of our synthesized videos is correlated with the number of example images provided at test time. The proposed attention mechanism can take advantage of a larger example set to better generate the network weights. (c) Visualization of attention maps when multiple example images are given. Note that when synthesizing the front of the target, the attention map indicates that the network utilizes more of the front example image, and vice versa.

at test time helps with the video synthesis performance. As shown in Figure 7(b), the result confirms our assumption.

**Limitations.** Although our network can, in principal, generalize to unseen domains, when the test domain is too different from the training domains it will not perform well. For example, when testing on CG characters which look very different from real-world people, the network will struggle. In addition, since our network is based on semantic estimations as input such as pose maps or segmentation maps, when these estimations fail our network will also likely fail.

## 5 Conclusion

We presented a few-shot video-to-video synthesis framework that can synthesize videos of unseen subjects or street scene styles at the test time. This was enabled by our novel adaptive network weight generation scheme, which dynamically determines the weights based on the example images. Experimental results showed that our method performs favorably against the competing methods.

## Footnotes

[1]For example, a segmentation mask or an image denoting a human pose.

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
