[Supplementary Material]

# Supplementary Material for Adaptive Video-to-Video Synthesis via Network Weight Generation

## 1 Details of the Intermediate Image Synthesis Network

As described in the main paper, our intermediate image synthesis network $H$ is based on the SPADE generator [2]. The SPADE generator contains two main components where one is a set of SPADE branches[1] and the other is the main image synthesis branch. A SPADE branch converts an input semantic image to a scalar map $\gamma$ and a bias map $\xi$, which are fed into a layer in the main image synthesis branch. Typically, multiple SPADE branches are used in the SPADE generator. In our adaptive video-to-video synthesis framework, we use the network weight generation module $E$ to dynamically generate the learnable weights in all the SPADE branches in the SPADE generator, as visualized in Figure 1.

## 2 Details of Our Strong Baselines

In the experiment section, we introduce three strong baselines for the adaptive video-to-video synthesis task. Here, we provide additional details of their architectures. Note that these baselines are only designed for dealing with one example image, and we compare the performance of the proposed method with these baselines on the one example image setting.

### 2.1 The Encoder Baseline

As discussed in the main paper, the Encoder baseline consists of an image encoder that encodes the example image to a style latent code, which is then directly fed into the head of the main image synthesis branch in the SPADE generator. We visualize the architecture of the Encoder baseline in Figure 2(a).

### 2.2 The ConcatStyle Baseline

As visualized in Figure 2(b), in the ConcatStyle baseline, we also employ an image encoder to encode the example image to a style latent code. Now, instead of feeding the style code into the head of the main image synthesis branch, we concatenate the style code with the input semantic image via a broadcasting operation. The concatenation is the new input semantic image to the SPADE modules.

### 2.3 The AdaIN Baseline

In this baseline, we use the AdaIN [1] for adaptive video-to-video synthesis. Specifically, we use an image encoder to encode the example image to a latent vector and use multilayer percetrons to convert the latent vector to the mean and variance vectors for the AdaIN operations. The AdaIN

Figure 1: Details of our intermediate image synthesis network. (a) The figure visualizes that our network weight generation network $E$ converts the set of input example images and their corresponding semantic images to the network weights for a SPADE module (or a spatial modulation module) in the SPADE generator. (b) The figure visualizes that our network weight generation network $E$ generates the weights for all the SPADE modules in the SPADE generator.

Figure 2: Details of the strong baselines. (a) The Encoder baseline tries to input the style information of the unseen domain by feeding a style code to the head of the main image synthesis branch in the SPADE generator. (b) The ConcatStyle baseline tries to input the style information of the unseen domain by concatenating the style code with the input semantic image, which are then inputted to the SPADE modules. (c) The AdaIN baseline tries to input the style information of the unseen domain by using AdaIN modulation.

parameters are fed into each layer of the main image synthesis branch. Specifically, we add an AdaIN normalization layer after a SPADE normalization layer as shown in Figure 2(c).

## Footnotes

[1]A SPADE branch is also referred to as a spatial modulation branch.