[Reviews · NeurIPS 2019]

Reviewer 1



This is a solid work that extended vid2vid to adapt examples from unseen domains. The key idea is the weight generation module that dynamically generates the weights of the network to transfer input examples from a new, unseen domain. While dynamic weight generation has been applied to a broad range of problems, this is the first work to address the vid2vid problem. The main downside of this work is the lack of details for the weight generation module: 1) If I understand correctly, the way the "style" of the new images is encoded in the beta vectors. The attention module compares the similarity of the poses in the testing examples to the current pose to be translated and produces a weight for combing the beta vectors (the new style from an unseen domain). It seems such an attention scheme is the key to outperform the alternatives. Please clarify if this is the case. 2) How It is not clear to me how many examples (the K parameter) is needed for training and testing. If attention is the key, then the more examples, the better. Does the method fail when K falls below a certain value? 3) It is not clear the comparison to the baselines are fair. It seems that all baselines use a single image to encode the "style" code or compute the parameters in AdaIN, whereas the proposed method uses K images. When it is true that the proposed method is unique in being able to fuse multiple reference images using attention, it is necessary to discuss this clearly in the paper. 4) It is also not clear that weight generation is necessary. Why not simply feed the weighted average appearance presentation (Line 154) to the generator, for example, via AdaIN layers. What is really the advantage of E_c here? Overall, the work addressed a very important problem and provided a novel solution. Although there are still artifacts, the results are very convincing from the perspective that shows clear improvement over the existing alternatives. The reason for not giving higher score is the lack of details and in-depth discussion for the network weight generation module, especially what actually makes the networks work-- the attention or the weight generation.

Reviewer 2



I would have liked to see more analysis of why the proposed "adaptation" approach helps to improve the quality of the results. Compared to the baselines and other methods, the empirical results in this paper are of a higher quality both quantitatively and qualitatively. It would have been interesting to show some failure modes and analyze why the failures happened - e.g. stretching the adaptivity to the limit. I don't find the notation in the paper very clear - it requires lots of rechecking to figure out what the different letters mean. If the paper is accepted, the authors should consider modifying their notation.

Reviewer 3



SUMMARY The paper tackles a few-shot generalization problem for video-to-video translation, where the goal is to synthesize videos of an unseen domain using few example images of the target domain. To this end the authors extend vid2vid with two modifications. First, they replace the image synthesis module with the recently proposed SPADE [41]. Second, they introduce an adaptive weight generation module which provides weight parameters to the SPADE model. This new module is implemented with a soft-attention mechanism to aggregate statistics from a few example images of the target domain. The proposed approach is evaluated on two tasks, motion retargeting and street scene style transfer. The authors compare with three baselines (implemented by the authors) as well as two existing work [1, 46]. The authors report both quantitative and qualitative results to demonstrate their approach. ORIGINALITY - The proposed adaptive weight generation module could be considered new. The idea of using a soft-attention mechanism for dynamic weight generation -- in particular, using the embedding a_t from the current input semantic image s_t as a key vector to retrieve relevant appearance statistics from example images e_{1:K} -- is intuitive and reasonable. Although attention-based encoding of image sets is not new, it is well used in this work. - It is unclear why the soft-attention based weight generation was necessary instead of, e.g., the AdaIN-based approach, as was done in FUNIT [31]. This needs to be better motivated -- Is AdaIN not applicable to the proposed approach, and if so, why? Why is it necessary to propose the attention-based method when existing approaches have already demonstrated the effectiveness of AdaIN in a similar setup (for images)? QUALITY - The quantitative results clearly show improvement over the compared baselines. The results on motion retargeting is particularly good, both qualitatively and quantitatively. - The authors do not discuss limitations/weaknesses of their method. It would've been nice if there was a discussion on this. CLARITY - The paper reads well overall. But some of important details are missing (see my comments below). SIGNIFICANCE - As mentioned above, the problem is of wide interest to this community and the proposed technique generalizes to different domains. The visual quality on motion retargeting is promising, and the code will be released. Given these, I think this work will likely to have a reasonable impact and could encourage follow ups.

[Author Response · NeurIPS 2019]

We thank all reviewers for their positive and constructive comments, such as the application important, the results impressive,
and the method generalizable. Below we first address the common questions, and then questions by individual reviewers.

(a) weight generation scheme   (b) modified SPADE
**Fig A: Our network architecture**

**Fig B: Original SPADE**

**Fig. C: Example attn maps**

**Fig. D: Comparison to AdaIN with varying dataset sizes.**

**Details of network architecture.**   We are sorry there were not enough
details provided in the paper due to the page limit. In Fig. A, we show
the network architecture when only one example image is used (i.e.
without the attention mechanism). Compared to the original SPADE
generator (Fig. B), we made two major modifications. First, instead of
generating the affine parameters at each layer independently, we reuse features from the previous layer when generating
parameters in the next layer (Fig. A(b)). Second, the weights in SPADE are determined on the fly (Fig. A(a)). After we
encode the example image, we apply adaptive pooling (AdaPool) to make it fixed spatial size. The results are fed to two
fully connected layers to generate the weights, which are used in corresponding convolutions (denoted by different colors).

| | AdaIN (avg) | AdaIN (attn) | Ours (attn) |
|---|---|---|---|
| FID | 129.90 | 113.83 | *76.53* |

**Table A: Comparison to AdaIN** *(5 example images).*

| | Ours (1 exp) | Ours (50 exps) | Ours (50 exps + finetune) | vid2vid |
|---|---|---|---|---|
| FID | 56.99 | 51.10 | *43.04* | 47.19 |

**Table B. Comparison to vid2vid**

**Attention mechanism.**   We combine features from multiple example images before feeding them to the AdaPool layer in
Fig. A(a). The attention maps are soft spatial maps of size $\mathbb{R}^{K \times N \times N}$, where $K$ is the number of examples and $N$ is the
spatial dimension. The map determines that for each spatial location in the output $(x, y)$, which spatial location in which
example image $(k, x_k, y_k)$ carries most relevant information. For example, when example images include both front and
back of the target person, the attention maps can help capture corresponding body parts during synthesis (Fig. C).

**Comparison to AdaIN.**   In AdaIN, information from the example image is represented as a scaling vector and a biased
vector. This operation could be considered as a 1x1 convolution with a group size equal to the channel size. From this
perspective, AdaIN is a constrained case of the proposed weight generation scheme, since our scheme can generate a
convolutional kernel with group size equal to 1 and kernel size larger than 1x1. Moreover, the proposed scheme can be
easily combined with the SPADE module. Specifically, we use the proposed generation scheme to generate weights for the
SPADE layers, which in turn generate spatially adaptive de-modulation parameters. To justify the importance of weight
generation, we compare with AdaIN both using weighted average and our attention module (Table A). We also compare
with AdaIN when different dataset sizes are used. The assumption is that when the dataset is small, both methods are able
to catch the diversity in the dataset. However, as the dataset size grows larger, AdaIN starts to fail since the expressibility is
limited, as shown in Fig. D.

**Limitations.** Although our network can, in principal, generalize to unseen domains, when the test domain is too different
from the training domains it will not perform well. For example, when testing on CG characters which look very different
from real-world people, the network struggles. We will include several failure examples in the revised version.

**R1: How many examples (the K parameter) are needed?** Although we demonstrated that more example images are
helpful (Fig. 7b in the paper), our method can work well when $K = 1$. We note that comparisons and examples shown in
the paper are using $K = 1$ (i.e., without the attention mechanism).

**R1: Comparison to the baselines and importance of weight generation.** As explained above, for all comparisons our
method only uses a single example image, which is the same as the baselines, so the comparisons are fair. This shows that
weight generation ($E_c$) is useful for generating good quality results. We will make this clear in the revised version.

**R2: Novelty.** To our best knowledge, we are the first to show that weight generation can be used to solve the adaptive
video-to-video synthesis problem. As recognized by R3, this will likely have a wide impact since the topic is of wide
interest and our method is generalizable. We will also fix the notations in the revised version.

**R3: Comparisons to Zhou et al. 2019 and [7].** We will cite and discuss these papers. Note that both these methods still
require *different* models for different persons, which has the same drawback as vid2vid. For example, they typically need
minutes of training data and days of training time, while our method only needs one image and negligible time for weight
generation. Moreover, since code of these methods are not released, we show comparisons to vid2vid in Table B for a
specific person. We find that our model renders comparable results even when $K = 1$. Moreover, if we further finetune our
model based on the example images, we can achieve comparable or even better performance.

**R3: Dataset.** The dataset is collected by the authors. Different videos are randomly divided into training/test sets. Each
video is further divided into clips to ensure each clip only contains one person and does not contain any scene transition,
which in general results in 30-1000 frames per clip. We will release the dataset for facilitating research in the field.

**R3: Human evaluation.** For each pair of comparisons, we generate 100 clips, each of them viewed by 60 workers. Orders
are randomized. Hence, each user preference score is computed based on the 6000 evaluations. As reported in the main
paper, our preference scores are significantly higher than $0.5$. We will also add the citation to the segmentation network.

[Meta-Review · NeurIPS 2019]

In general, all the reviewers liked the results presented in the paper, and the topic is timely and important in video synthesis. There are concerns that comparison with some recent approaches on motion retargeting is missing and the details of the network generation module is missing. After the rebuttal, one of the reviewers increase the rating to accept. The AC strongly encourages the authors to incorporate the comments from the reviewers into the final version.